# ZDDP Tribofilm Formation from a Formulated Oil on Textured Cylinder Liners

Leonardo C. Dias [1], Giuseppe Pintaude [2], Alessandro A. O. F. Vittorino [3] and Henara L. Costa [1,3,*]

[1] Laboratory of Surface Engineering, School of Engineering, Universidade Federal do Rio Grande, Rio Grande 96203 900, Brazil; leonardocondedias@gmail.com
[2] Academic Department of Mechanics, Universidade Tecnológica Federal do Paraná, Curitiba 81280 340, Brazil; giuseppepintaude@gmail.com
[3] Laboratory of Tribology and Materials, Universidade Federal de Uberlândia, Uberlandia 38408 100, Brazil; alessandro_xam@hotmail.com
[*] Correspondence: henaracosta@furg.br

**Abstract:** Surface texturing can improve lubrication and entrap wear debris but increases the effective roughness of the surfaces, which can induce higher contact pressures. On the one hand, this can be detrimental, but on the other hand, the increase in contact pressure could be used to activate the formation of a ZDDP tribofilm from fully-formulated lubricants. This work investigates the synergistic effect between surface texturing via Maskless Electrochemical Texturing (MECT) and ZDDP additive. The surface texture consisted of an array of annular pockets manufactured on a gray cast iron cylinder liner. These textured surfaces were evaluated by scanning electron microscope (SEM) and energy-dispersive X-ray spectroscopy (EDX). The results indicated that surface texturing via MECT changes the chemical composition of the surfaces, by inducing a preferential dissolution of the metal matrix. Consequently, it exposed the carbon present in the material. The tribological performance was evaluated by a ring-on-cylinder-liner tribometer in reciprocating sliding under boundary lubrication conditions using both a base oil and a commercial formulated oil containing ZDDP additive. For comparison, a commercially honed liner was also tested. After the tribological tests, the surfaces were evaluated by white light interferometry and SEM/EDX. Although the textured surfaces showed higher friction, they induced more ZDDP-tribofilm formation than conventional cylinder liner finish.

**Keywords:** surface texturing; MECT; ZDDP; piston-liner; tribofilm

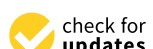



## 1. Introduction

For decades, internal combustion engines (ICEs) have used the control of surface topography to improve their tribological performance, which ultimately has resulted in reduced carbon emissions [1]. Among those, plateau honing has been used since the 1950s to combine good load support given by a smooth surface finish and lubricant supply conferred by cross-hatched grooves [2]. In the early 2000s, surface texturing was proposed as an advantageous technique for topographic modification and control of the piston-liner system [3].

In the piston-liner system, although full film lubrication prevails for most of the cycle/stroke, the very low velocities at the two extremes of the stroke (top dead center and bottom dead center) result in mixed and boundary lubrication [4]. Different mechanisms can improve performance in the presence of surface textures covering the different lubrication regimes present in the piston-liner system [5].

Effects of surface texturing for the piston-liner system have been studied using both analytical/numerical models and experiments [5–7]. Etsion's group carried out the first studies, which proposed an analytical model showing the benefits of partial texturing over full texturing and optimal performance for a textured portion of 60%, area density of 50%,

and aspect ratio between 0.04 to 0.1, corroborated by experiments [8], including engine tests [9]. Since then, a more accurate account of the cavitation in the textures has been used in the models [10]. In particular, modelling of surface texturing for the cylinder-liner system using 3D deterministic models that consider mass-conservative cavitation has been shown to be more realistic [11]. Experimentally, significant insight into the lubrication of textured liners has been provided by in-situ visualization of the contact using ultra-thin film optical interferometry and fluorescence, showing the importance of two less cited mechanisms in the literature: inlet suction [12,13] and shear area variation [14]. The use of engine tests has also been fundamental to give a realistic measurement of the real benefits of surface texturing in terms of wear, fuel consumption, and engine efficiency [15,16].

The literature is abundant in reviews that help to establish the state-of-the-art regarding the main effects of surface texturing under different lubrication regimes. Of particular importance are reviews regarding hydrodynamic lubrication of textured surfaces [7,17,18], since full film lubrication of a conformal contact prevails for most of the engine stroke. Reviews specifically for the effects of texturing in the piston-liner system highlight how texturing can influence friction in the full-film lubrication areas of the stroke, as well as shift the transition between full-film and mixed lubrication and affect the boundary lubrication at the ends of the stroke [5].

Different authors have investigated the effects of the shape of the texture specifically for the piston-liner system [19]. The main shapes investigated have been circular dimples [20–23], chevrons [24,25], and parallel or crossed grooves [16,20,22,26,27] using experiments and numerical simulation. However, the use of textures containing ring-like pockets has not been investigated yet.

Laser texturing is a very important technology for the piston-liner system [5], but presents some limitations in terms of texturing speed and cost for texturing relatively cheap components in industrial lines [28,29]. A recent review has established the current developments and trends in laser texturing technology [30]. Alternatively, Maskless Electrochemical Texturing (MECT) is a simple, cheap, and fast technique to texture conductive surfaces, although it is not a commercial technique yet [31]. MECT uses localized micro- electrochemical machining of an anodic surface against a patterned cathode, but the individual workpieces do not require previous masking [32] as an advantage over other electrochemical techniques, such as through-mask electrochemical micromachining (TMEMM) [33–35]. Despite some drawbacks of electrochemical texturing in general, such as the difficulty of controlling the technique, the complexity involved in producing the patterned cathode, and the strong dependence of the anodic dissolution on the process parameters [36], MECT has been successfully adapted to texture cylinder liners [37,38].

Formulated oils are fundamental for the transportation sector, since the base oil alone cannot confer either the necessary lubricity and low wear required for the lubricated components or the required stability of the lubricant under the extreme conditions found in the engine [39]. Among the anti-wear additives, zinc dialkyldithiophosphate (ZDDP) is undoubtedly the most successful, despite the environmental concerns regarding possible damage to the exhaust system when it leads to the formation of phosphated ashes [40]. In addition to its anti-wear properties, the ZDDP additive also acts as an anti-oxidant and anti-corrosive. In fact, the transportation sector is the largest world consumer of ZDDP additive [41].

ZDDP is dissolved in the lubricant, often at concentrations below 0.08%wt. For the ZDDP molecule (initially dissolved in the lubricant) to form a protective tribofilm, a series of tribochemical reactions need to occur when the surface asperities rub against each other. Ultimately, the formed ZDDP tribofilm consists mainly of zinc sulfide and Fe, Zn polyphosphates [42,43]. Different mechanisms have been suggested to induce the tribochemical reactions leading to this tribofilm, including flash temperatures at the contacting asperities, high pressures involved in the contact, triboemission and surface catalysis [44]. Important works in the literature review the main mechanisms and tribochemical reactions involved in forming protective anti-wear films from the ZDDP additive, as well as their detailed

morphology and chemistry [41,44,45]. Obviously, the formation of the ZDDP tribofilm requires the occurrence of boundary lubricant, but there are fewer studies related to the effects of surface texturing under boundary lubrication conditions [46,47].

Among the mechanisms suggested in the literature, the most compelling evidence favors the hypothesis of high contact pressures. Zhang and Spikes [44] have experimentally proven that the tribochemical reactions needed to form the ZDDP tribofilm are induced by the shear stresses acting at the contact interface. Surface texturing produces rougher surfaces [48] and, therefore, should induce higher stresses at the contact. Based on this premise, it is reasonable to suppose that surface texturing might help induce the tribochemical reactions involved in the formation of ZDDP tribofilms. Hsu, et al. [49] used a roller bearing apparatus to verify the interactions between surface texturing by direct laser interference patterning (DLIP) and ZDDP additive. Their results suggested that a cross-like pattern of hemispherical protuberances and grooves could help induce ZDDP activation because Raman analysis showed the presence of Zn, Fe sulphides on the top of the hemispherical protuberances (the phosphates are not Raman-sensitive).

Similarly, for a titanium alloy (Ti6Al4V) textured by DLIP, EDX of a cross-section of the wear scar indicated the presence of Zn after reciprocating tests using a formulated oil containing ZDDP. The authors suggested that the confinement of the lubricant in the grooves helped induce tribofilm formation [50]. For another boundary additive, Molybdenum Dithiocarbamate (*MoDTC*), which is responsible for reducing friction in the boundary regime, Khaemba, Azam, See, Neville and Salehi [47] pointed out a possible correlation between the high stresses at the contact for textured surfaces and the activation of tribochemical reactions responsible for the formation of a $MoS_2$ tribofilm.

In the present work, we aimed to shed further light on the interaction between surface texturing and ZDDP activation. For that, we compared textured liners and commercially-honed liners, since honing is currently the most important technology for the surface finish of liners [4]. The textures tested (pockets with ring-like geometry) aimed to combine a good load-bearing capacity in the middle of the rings with the high contact pressure at the edges of the pockets. Such textures have not been tested yet for the piston-ring system, to the best of our knowledge. Another novel aspect is the use of textures produced by MECT to interact with ZDDP, which are very cheap and fast to produce, therefore with great potential to be industrially used in the piston-liner system. For that, we used reciprocating sliding of ring sections sliding against a textured liner, which should reproduce very well the conditions in car engines. Thus, the main novelty of the work is to test if a simple and cheap surface texturing technique can improve the activation of anti-wear additives cylinder liners (which are cheap components), with potential to improve their lifetime. Most of the work for cylinder liners has been focused on reducing friction.

## 2. Materials and Methods

### 2.1. Materials

The specimens were cut from commercial gray cast iron cylinder liners; the external diameter of the liner was 131 mm. The dimensions of the sections after cutting are shown in Figure 1a. The microstructure of the specimens after polishing and etching with Nital 3% is presented in Figure 1b, showing type A graphite flakes (according to ASTM A247 standard) in a pearlitic matrix. Our group has previously carried out a thorough analysis of this material in terms of mechanical properties and microstructure using optical microscopy and SEM/EDS [51]. The micro hardness of the pearlitic matrix is $328 \pm 17$ $HV_{0.1}$. A total of twelve specimens were prepared for the tests, six using the original surface condition of the liners (plateau honing) and six for the textured condition.

The counterbodies were segments of commercial compression piston rings (asymmetrical barrel shape) made of martensitic stainless steel with a CrN PVD coating. The ring elastic modulus is 220 GPa, and the coating microhardness is $1800 \pm 296.1$ $HV_{0.05}$. Further details regarding the mechanical, topographical and microstructural analysis of the rings can be found in [52].

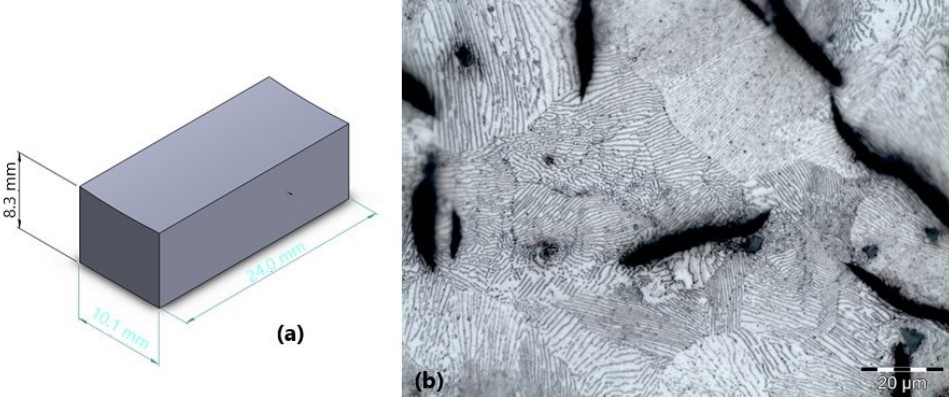

**Figure 1.** Cylinder liner specimens: (**a**) specimen dimension after cutting; (**b**) microstructure after etching with Nital 5%.

The lubricants used in this work were a pure polyalphaolefin base oil with a kinematic viscosity of 4 cSt (PAO 4) without any additives and a commercial monograde diesel formulated oil SAE30 CF (Petronas Urania) containing ZDDP additive. The main rheological properties of both lubricants are summarized in Table 1.

**Table 1.** Rheological properties of the lubricants.

| Lubricant | Density 15 °C | Kinematic Viscosity 40 °C | Kinematic Viscosity 100 °C |
|---|---|---|---|
| PAO 4 | 0.82 | 18 cSt | 4 cSt |
| SAE 30 | 0.881 | 92 cSt | 11 cSt |

## 2.2. Methods

### 2.2.1. Surface Characterization

The 3D surface topography of the liners (honed and textured) was assessed both before and after the wear tests using a white light interferometer model CCI Talysurf Lite. The measured areas were 0.83 mm × 0.83 mm with a number of measured points of 1024 × 1024. For statistical repeatability, three areas of 0.83 mm × 0.83 mm were assessed per specimen condition. The topographic measurements were analyzed using the software Mountainsmap version 9.1. To calculate the topographic parameters, the areas measured were first levelled. Then, the form was removed due to the cylindrical shape of the liners using a second-order polynomial function, as exemplified in Figure 2. Note that in Figure 2 measurement areas larger than 0.83 mm × 0.83 mm were used to enable a broader visualization of the surface morphologies. For the textured liners it was observed that the shape removal also removed some of the textures (Figure 2c), but the textures were still clearly visible on the remaining surface (Figure 2b). This small error was necessary to ensure that the cylindrical shape of the liners did not influence the roughness parameters. The parameters chosen to characterize the topography of the liners and the topographical changes due to wear in the lubricated tests were *Sq* (RMS roughness, which accounts for the average height of the asperities), *Ssk* and *Sku* (skewness and kurtosis of the height distribution curve, respectively, which account for the asymmetry and peakedness of the curve distribution of the asperity heights, respectively) and the functional parameters of the *k* series (*Spk*, *Sk* and *Svk*), calculated from the Abbot-Firestone curve [53]. *Spk* is associated with the highest asperities, which tend to be worn-out first; *Sk* is associated with the core of the topography; and *Svk* with the deepest valleys, responsible for lubricant storage. Additionally, the surface roughness of the commercial piston rings was assessed before the tests.

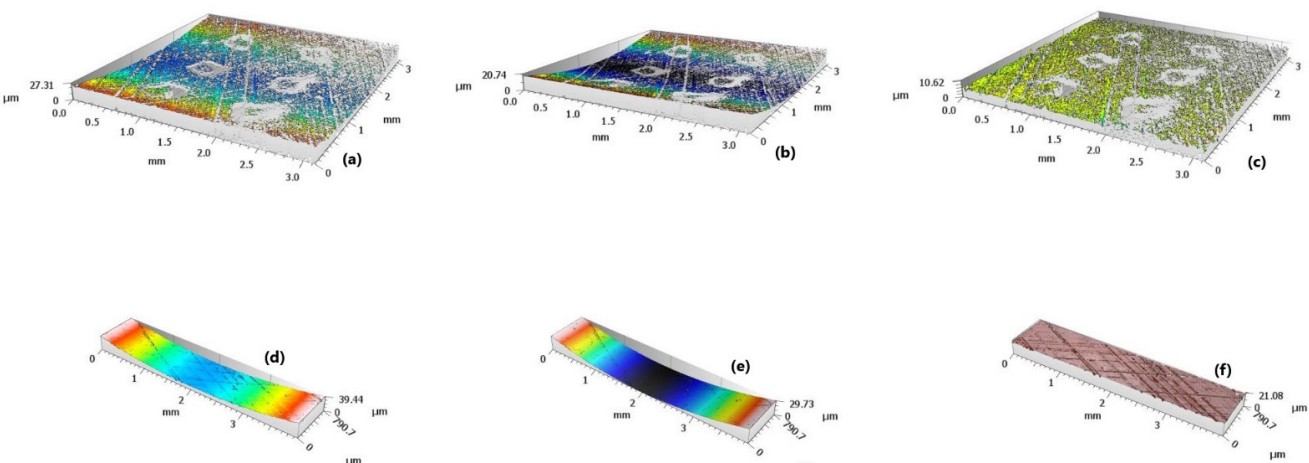

**Figure 2.** Methodology for form removal: (**a**) textured liner; (**b**) textured liner, roughness; (**c**) textured liner, form alone; (**d**) honed liner; (**e**) honed liner, roughness; (**f**) honed liner, form alone.

2.2.2. Surface Texturing of the Liners

Maskless electrochemical texturing (MECT) was used to texture the cylinder liners. The electrolyte flux was horizontal, parallel to the electrode faces, as proposed in [54] to increase the efficiency of the technique. The MECT chamber is shown schematically in Figure 3a,b. It consists of a hollow cylindrical body made in nylon (electrical insulator) and divided into a front part (6) and a rear part (5), carefully sealed using sealing rings. The electrolyte (pumped in the chamber) enters through the hole (4), flowing horizontally between the cathode and the anodic workpiece, and leaves through the hole (3). The metallic rods (6) and (7) are responsible for the electric connection of the cathode (1) and the anodic workpiece (2) with a power supply. A central fuse controls the movement of the rear part so that the gap between the cathode and anode can be adjusted with the help of a micrometric dial gauge. The texturing chamber is very versatile in terms of the geometry of the anodic workpiece. As shown in the detail presented in Figure 3c, an internal insert fixed to the rear part of the chamber (and the moving fuse) is used to hold the workpiece. This insert can be easily interchanged if the geometry of the workpiece changes just by machining a different insert.

Similarly, an insert is fixed to the front part of the chamber that holds the cathode. In this work, the cathode was machined from an AISI 316 stainless steel block (Figure 3d). One of the faces of the cathode is convex, with an external diameter of 131 mm (corresponding to the internal diameter of the liners). The cathode was then masked using an adhesive vinyl film previously patterned by laser (Figure 3e). Although MECT requires masking of the cathode, it is carried out only once, and then the cathode can be used to texture innumerous anodic workpieces, without the need for individual masking of each workpiece. Further details about this newly developed MECT chamber can be found in [54].

The power supply provides pulsed current, which helps to cool down and remove the anodic dissolution products from the gap between the cathode and anode during the off-time of the cycle. The electrolyte was a 2 M NaCl + 1 M NaNO$_3$ solution in distilled water. The processing parameters used during MECT were selected based on previous works [54] and are summarized in Table 2. After texturing, the specimens were carefully rinsed and then dried with isopropanol.

**Table 2.** MECT texturing parameters.

| Gap (μm) | Voltage (V) | Electrolyte Flow (L.s$^{-1}$) | Duty Cycle $t_{on}/(t_{off} + t_{on}) \times 100\%$ | Frequency (Hz) | Time (s) |
|---|---|---|---|---|---|
| 300 | 25 | 0.163 | 20% | 50 | 300 |

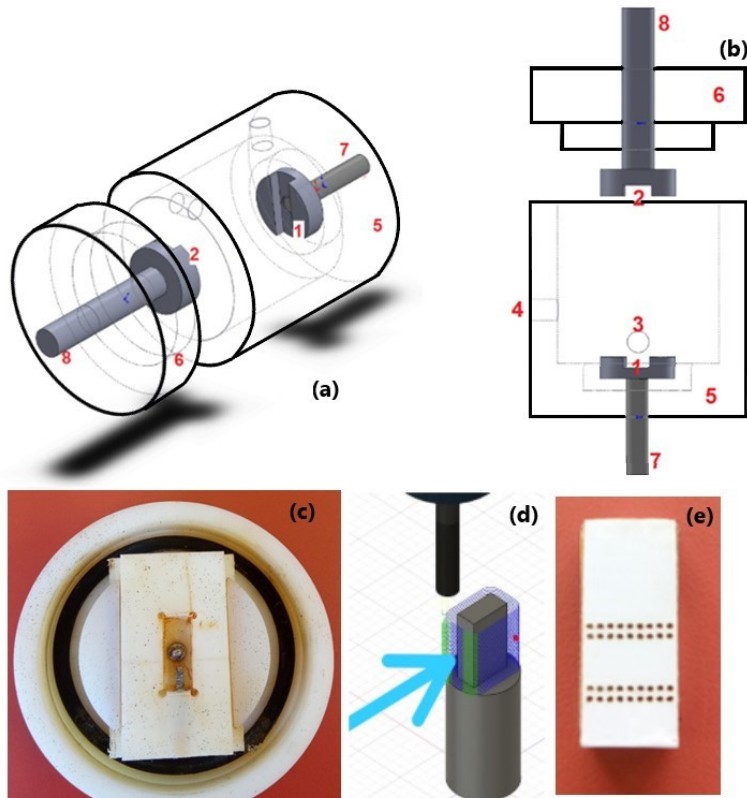

**Figure 3.** MECT apparatus: (**a**) isometric view of the prototype; (**b**) lateral view; (**c**) detail of the specimen holder; (**d**) machining of the cathode; (**e**) masked cathode.

### 2.2.3. Tribological Tests

The tribological performance of the specimens was evaluated using a universal tribometer model CETR-UMT—Bruker under lubricated reciprocating sliding. The immersion lubricant container can heat 20 mL of lubricant using resistors. The contact comprises ring segments sliding against a specimen cut from a cylinder liner. The combinations of specimens/lubricants tested are summarized in Figure 4. Three repetitions were carried out for each combination to ensure repeatability of the results.

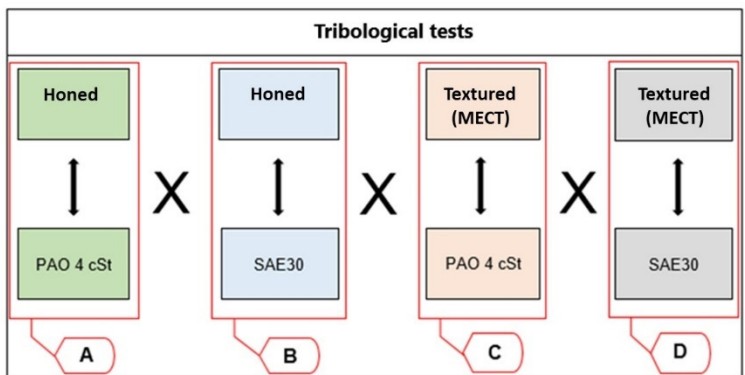

**Figure 4.** Conditions tested—A: honed with PAO; B: honed with SAE 30; C: MECT with PAO; D: MECT with SAE 30.

Two types of tests were carried out: i. short duration; ii. long duration. The short duration tests (120 s) aimed to vary the test frequency to measure the friction coefficient at different average speeds. The reciprocating frequencies were 1, 2.5, 5, 7.5, and 10 Hz. These tests allowed us to assess how friction varied with speed in order to identify in

which region of the Stribeck curve the tests were carried out (boundary, mixed or full film lubrication) considering the average stroke speed. Values of friction force, normal force, and relative position between specimen and counterbody were measured at an acquisition rate of 1 kHz. The long duration tests (3 h) aimed to evaluate the wear of the specimens, as well as monitor friction changes throughout the tests. The reciprocating frequency was 1 Hz, so the average speed was low, ensuring the predominance of boundary lubrication. This is essential to allow the activation of the ZDDP anti-wear additive present in the formulated oil. Due to many data, the acquisition rate was reduced to 0.1 kHz in the long duration tests.

Before the tribological tests, the rings and liners were cleaned with acetone, followed by ultrasonic cleaning in isopropanol and drying. In order to ensure good alignment between the ring and the liner segments, the liner surface was painted with non-permanent ink. Before starting the tests, the alignment was checked by manually moving the ring over the liner (without load applied) until the ink was uniformly removed from the liner, generating a uniform contact area. It must be emphasized that the short and long duration tests were carried out in sequence, without interruption between them.

After the tests, the specimens were cleaned, dried, and analyzed via white light interferometry and scanning electron microscopy (SEM, equipment model Carl Zeiss EVO MA 15 under 20 kV) coupled to energy-dispersive X-ray spectroscopy (EDS) to investigate the wear mechanisms.

### 3. Results

*3.1. Characterization of the Liners and Rings*

The topographical parameters of the rings (after form removal) are presented in Table 3. The asymmetrical barrel-shape of the rings was confirmed by selecting a profile from the 3D measurement (Figure 5). These values corroborate the results of an extensive topographic characterization of asymmetrical barrel-shape rings in the literature [55].

**Table 3.** Roughness parameters of the commercial piston rings used in the tests.

| *Sq* (μm) | *Ssk* | *Sku* | *Spk* (μm) | *Sk* (μm) | *Svk* (μm) |
|-----------|-------|-------|------------|-----------|------------|
| 1.035 | −0.003 | 7.406 | 1.451 | 0.7349 | 1.289 |

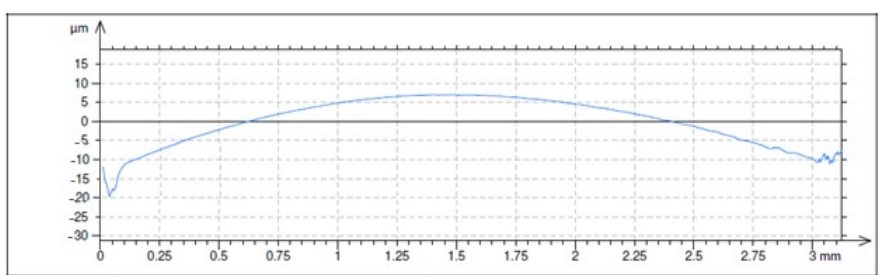

**Figure 5.** Profile of the rings.

The characterization of the morphology of the liners is exemplified in Figure 6. Using the software Mountainsmap, the angle of the cross-hatched grooves in the honed liners was measured as approximately 53° (Figure 6b). The textured liners showed annular pockets distributed over the previously honed surface (Figure 6c). The average external diameter of the rings was 790 μm, and the internal diameter was 600 μm, with standard deviations of 270 μm and 190 μm, respectively. The average spacing between pockets was 420 μm, with a standard deviation of 220 μm. The 3D map of an individual pocket after form removal is shown in Figure 6d, confirming that the internal portion of the rings was not machined. The selected profile of the pocket presented in Figure 6e shows a pocket depth of approximately 4 μm.

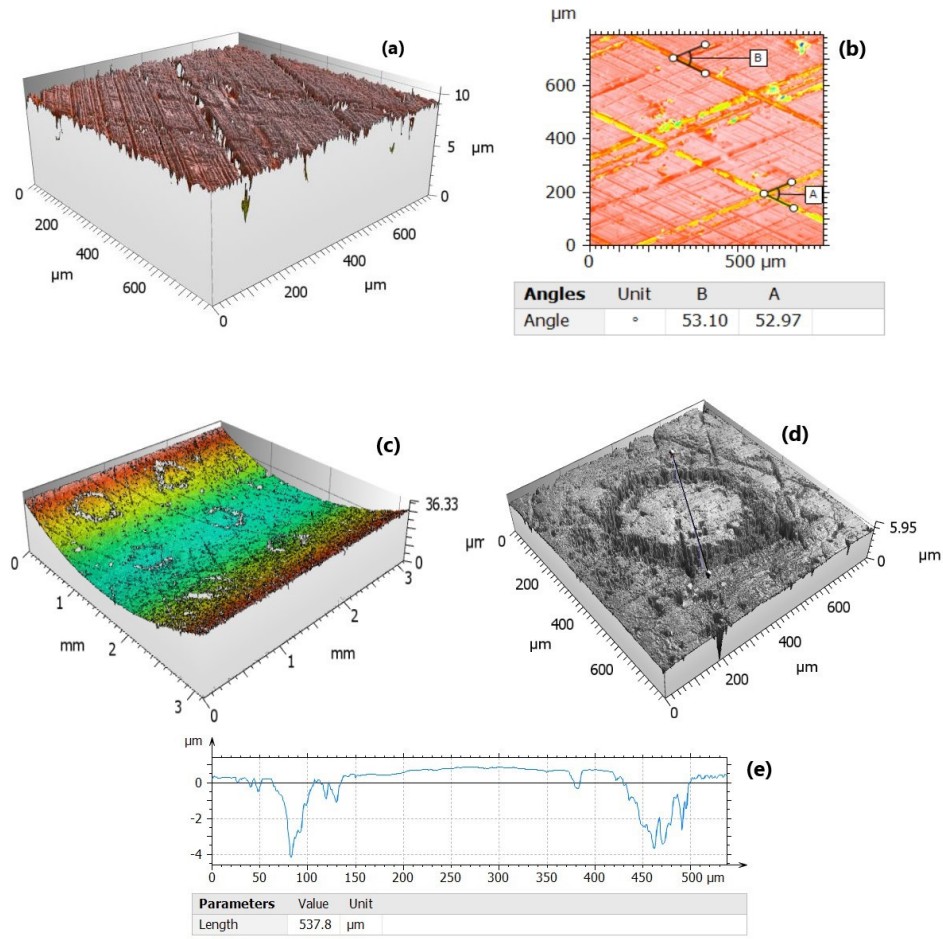

**Figure 6.** Typical topography of the liners: (**a**) honed, 3D map; (**b**) honed, top view; (**c**) textured, 3D map; (**d**) individual pocket, 3D map; (**e**) individual pocket, selected profile.

Calculations using the measurement of the pockets showed a ring width of 95 μm. The area coverage was 33.5% considering the total pocket (external diameter), but since the internal area of ring showed the same height as the areas outside the pockets, the actual area coverage (including the internal area of the rings) was thus 24%. The aspect ratio between the pocket depth and the ring width was 0.04. The aspect ratio of 0.04 is in the range optimized by many studies in the literature, since large aspect ratios can lead to starvation in the contact [8]. The area coverage of around 24% has shown good tribological results under hydrodynamic lubrication using both numerical simulations and experiments [56]. The choice of external and internal diameters of the rings was limited by the MECT technology, but it must be emphasized that current studies in our group are developing other masking techniques for the cathode that enable the production of substantially smaller diameters, which should be available in the near future.

A representative annular pocket of the textured liners was evaluated by SEM/EDX. Chemical contrast in the ring region was found in the backscattered electron (BSE) image (Figure 7a), where the ring region is darker than the other areas. For the backscattered electrons the regions containing atoms with lower atomic number tend to scatter less the incident beam because their nucleus is smaller, therefore appearing darker. Considering the nature of MECT, it is probably related to oxidation in the region of the rings. The EDS analysis comparing the region of the rings (1) with the regions (2) and (3) confirms this hypothesis (Figure 7b). Moreover, EDS shows relatively less iron and more carbon in the region of the rings (1), suggesting preferential dissolution of the iron matrix in relation to the graphite, as already reported in the literature for gray cast iron [38]. Finally, residual Na and Cl used in the electrolyte were detected in the pockets.

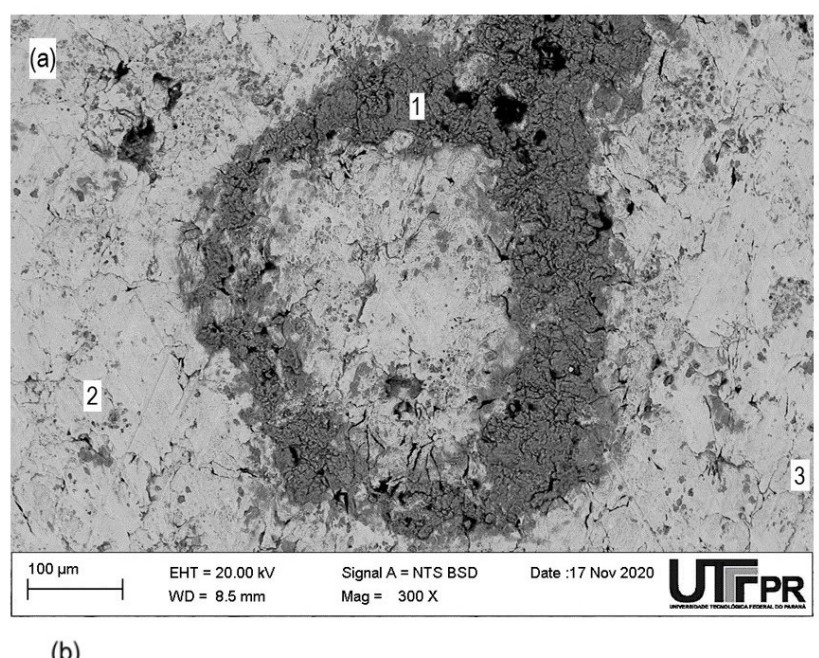

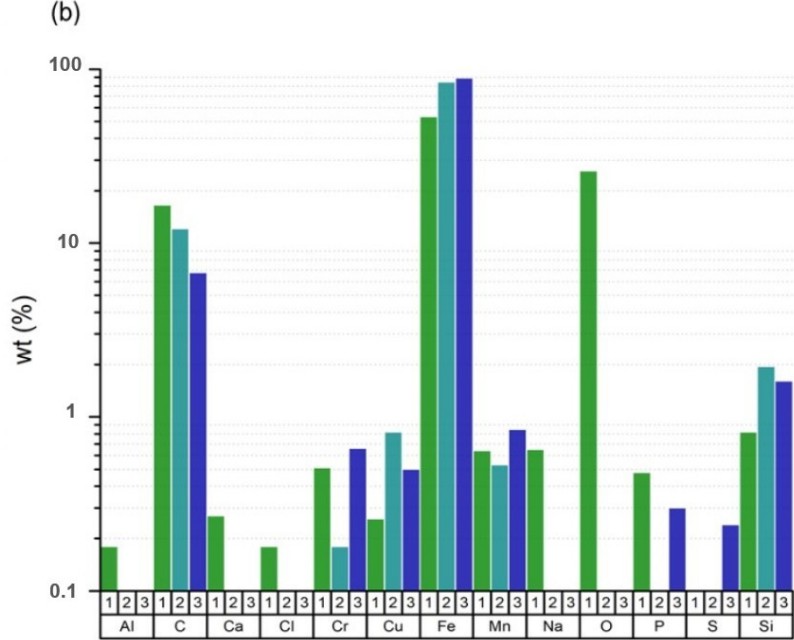

**Figure 7.** Typical annular pocket in the textured liner: (**a**) BSE image; (**b**) EDS analysis.

### 3.2. Short Duration Tribological Tests

The short duration tests at varying reciprocating frequencies aimed to verify the region of the Stribeck curves that corresponded to each average speed (corresponding to each reciprocating frequency). The variation of the average friction coefficient with the reciprocating frequency is shown in Figure 8, which indirectly represented a Stribeck curve. The raw COF values (without any previous averaging by the test rig software) were averaged along 2 min of test at each frequency. For the tests with the formulated oil (B—honed and D—textured), friction values varied little with the average speed, although a slight tendency for friction reduction was observed for the highest frequencies. A similar behavior was observed for the base oil when the liner was honed (curve A). A large scattering of the friction values was observed for the base oil with a textured liner. This suggests that the friction values increase substantially at the end of the strokes, where the instant speed is low, due to the lack of boundary additives. However, for all the tested conditions, on average the lubrication regime was boundary along the largest portion of

the cycles, particularly for the lower frequencies. For the highest frequencies, a transition probably started towards mixed lubrication, so that friction showed a slight tendency to reduce with average speed. Therefore, for the long duration tests, aiming to investigate wear for the different surface/lubricant conditions, the lowest frequency of 1 Hz was selected. The literature shows that most boundary additives require asperity contact and high shear stresses at the contact interface for their activation [44]. To confirm the prevalence of boundary lubrication, values of the lambda parameter ($\lambda$) were calculated as the ratio between the minimum lubricant film thickness ($h_{min}$) and the combined RMS roughness of the surfaces ($\sigma^*$). The calculation of $\sigma^*$ for the textured surfaces used the $Sq$ values for the entire surface, including the annular pockets. The calculation of $h_{min}$ used Equation (1), which is the Hamrock and Dowson´s equation to estimate the minimum film thickness for a line contact [57], considering that the barrel-shape of the ring confers an approximately cylinder-on-flat geometry for the contact with the liner.

$$\frac{h_{min}}{R} = 1.657\left(\frac{U\eta\alpha}{R}\right)^{0.7273}\left(\frac{W}{LRE^*}\right)^{-0.0909} \tag{1}$$

where $R$ = ring radius (proprietary), $U$ = average speed for each frequency, $\eta$ = lubricant viscosity, $\alpha$ = lubricant piezoviscous coefficient (estimated as $1.1 \times 10^{-8}$ Pa$^{-1}$ using the Barus equation [58]), $W$ = normal load (170 N), $L$ = contact length (10 mm) and $E^*$ = combined elastic modulus of the two contacting surfaces (calculated as 84.4 GPa). The calculated values presented in Table 4 confirm $\lambda << 1$, showing the prevalence of boundary lubrication conditions indeed. However, it should be emphasized that the values presented in Table 1 used the minimum film thickness calculated using an equation that considers smooth surfaces. The receding features present in the liners (grooves for the honed liners and pockets for the textured liners) should increase the actual $\lambda$ values. This reinforces our choice of using the lowest frequency (1 Hz) for the long duration tests, so that even with this possible increase, $\lambda$ remains lower than 1. Boundary lubrication is a necessary condition to activate the tribochemical reactions leading to the formation of the ZDDP tribofilm.

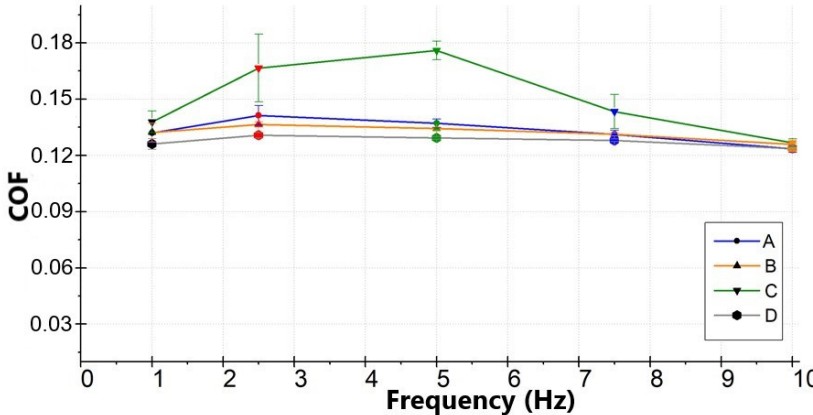

**Figure 8.** Average COF against reciprocating frequency (Stribeck curve) for the different specimen/lubricant combinations.

**Table 4.** Calculated $\lambda$ ($h_{min}/\sigma^*$) for the different test conditions.

| Frequency | A (Honed PAO) | B (Honed Formulated) | C (Textured PAO) | D (Textured Formulated) |
|---|---|---|---|---|
| 1 Hz | 0.013 | 0.052 | 0.004 | 0.004 |
| 2.5 Hz | 0.025 | 0.102 | 0.009 | 0.007 |
| 5 Hz | 0.041 | 0.168 | 0.014 | 0.012 |
| 7.5 Hz | 0.054 | 0.226 | 0.019 | 0.016 |
| 10 Hz | 0.067 | 0.280 | 0.024 | 0.020 |

### 3.3. Long Duration Tribological Tests

The evolution of the coefficient of friction (COF) with the number of reciprocating cycles for the long duration tests (3 h) is shown in Figure 9. Since the number of points acquired was huge, the graphs were plotted in intervals of 20 min each, otherwise the computation would become difficult. The average values in each cycle are shown by highlighted darker lines, whereas the cloud of points in slightly lighter shades shows all the raw COF values acquired at each point within the cycles. As expected, the tests with the base oil (curves A and C) showed substantially higher friction and scattering of the values. This occurs because the lack of boundary additives in the base oil makes friction very high at the ends of the strokes, where the speed virtually goes to zero before the change in movement direction (therefore almost no EHL liquid film), although in the regions of the cycle with high speed an EHL film may separate the surfaces. For the formulated oil (curves B and D), friction was lower and stable, with the cloud of lighter points almost imperceptible when compared with the base oil, confirming the action of the boundary additives that reduce friction when the speed is low.

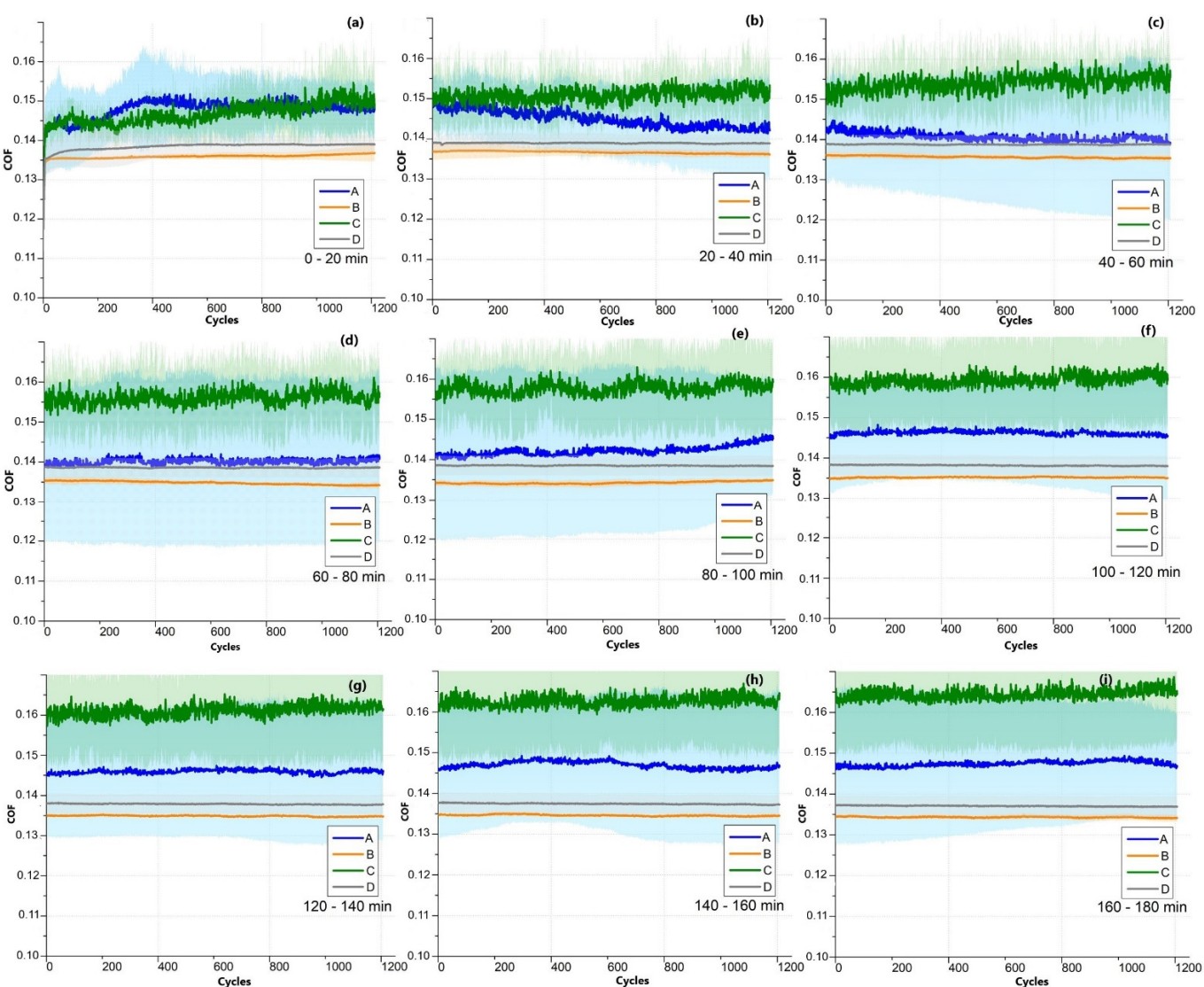

**Figure 9.** Evolution of COF with the number of cycles during subsequent test intervals: (**a**) 0 to 20 min; (**b**) 20 to 40 min; (**c**) 40 to 60 min; (**d**) 60 to 80 min; (**e**) 80 to 100 min; (**f**) 100 to 120 min; (**g**) 120 to 140 min; (**h**) 140 to 160 min; (**i**) 160 to 180 min.

For the initial 20 min of the test in base oil (Figure 9a), the honed and textured liners showed statistically similar behavior, but as the test time increased the friction values of the textured liner tended to increase slightly (from around 0.145 in Figure 9a to around 0.165 in Figure 9f), which did not happen for the honed liner. For the tests with the formulated oil (curves b and d), friction remained stable and low for both the honed and textured liners, not increasing with test time, confirming that the tribofilms formed on the surfaces protected them against severe metal asperity contact. However, the friction values were consistently slightly higher for textured liner when compared with the honed liner throughout the whole test duration.

*3.4. Wear Analysis*

First the combined effect of lubricant and surface finish on wear and tribofilm formation was analyzed from a qualitative point of view. Topographic images of the interfaces between the wear track and the unworn surface are presented in Figure 10. In the 3D maps (Figure 10a,b), the blue lines delimitate the areas within and outside the wear tracks. It can be clearly seen that for the base oil (Figure 10a) the surface appears worn-out within the wear track, with much less visible pockets. The selected profile (Figure 10c), indicated by the semitransparent plane in the 3D map, shows a clear height difference between the wear track (left portion) and the unworn region (right portion). For the formulated oil (Figure 10b), wear is negligible in the wear track; instead, the formation of a tribofilm is evidenced, which seems to concentrate in the areas close to the annular pockets. The selected profile (Figure 10d) shows that the wear track appears slightly higher (right portion) than the unworn area (left portion).

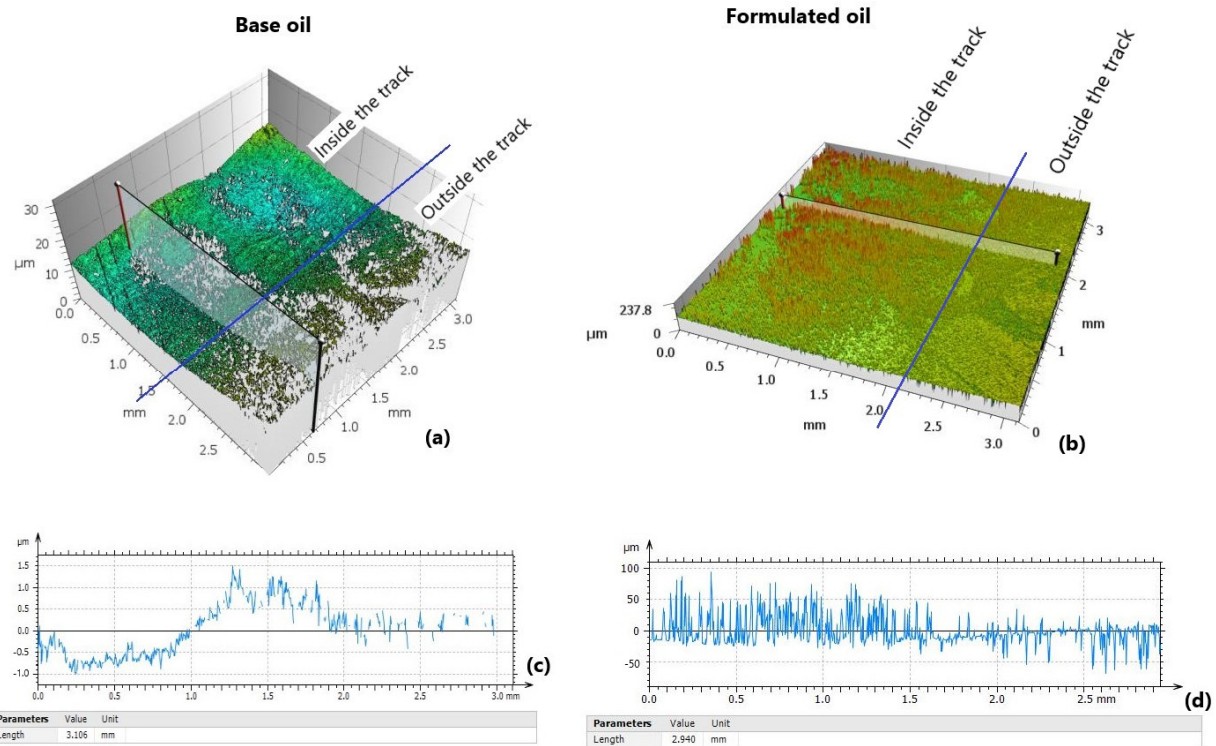

**Figure 10.** Topographic analysis of the textured liner after the reciprocating tests: (**a**) 3D map, base oil; (**b**) 3D map, formulated oil; (**c**) 2D selected profile, base oil; (**d**) 2D selected profile, formulated oil.

The worn interface was less visible for the honed liners, so 3D maps of regions inside and outside of the wear tracks are presented separately. For the base oil, the wear inside the track (Figure 11b) removed the honing grooves substantially, as well as produced damage in the plateau regions when compared with the region outside the track (Figure 11a). For

the formulated oil, the regions inside and outside the worn tracks showed very similar morphologies (compare Figure 11c,d).

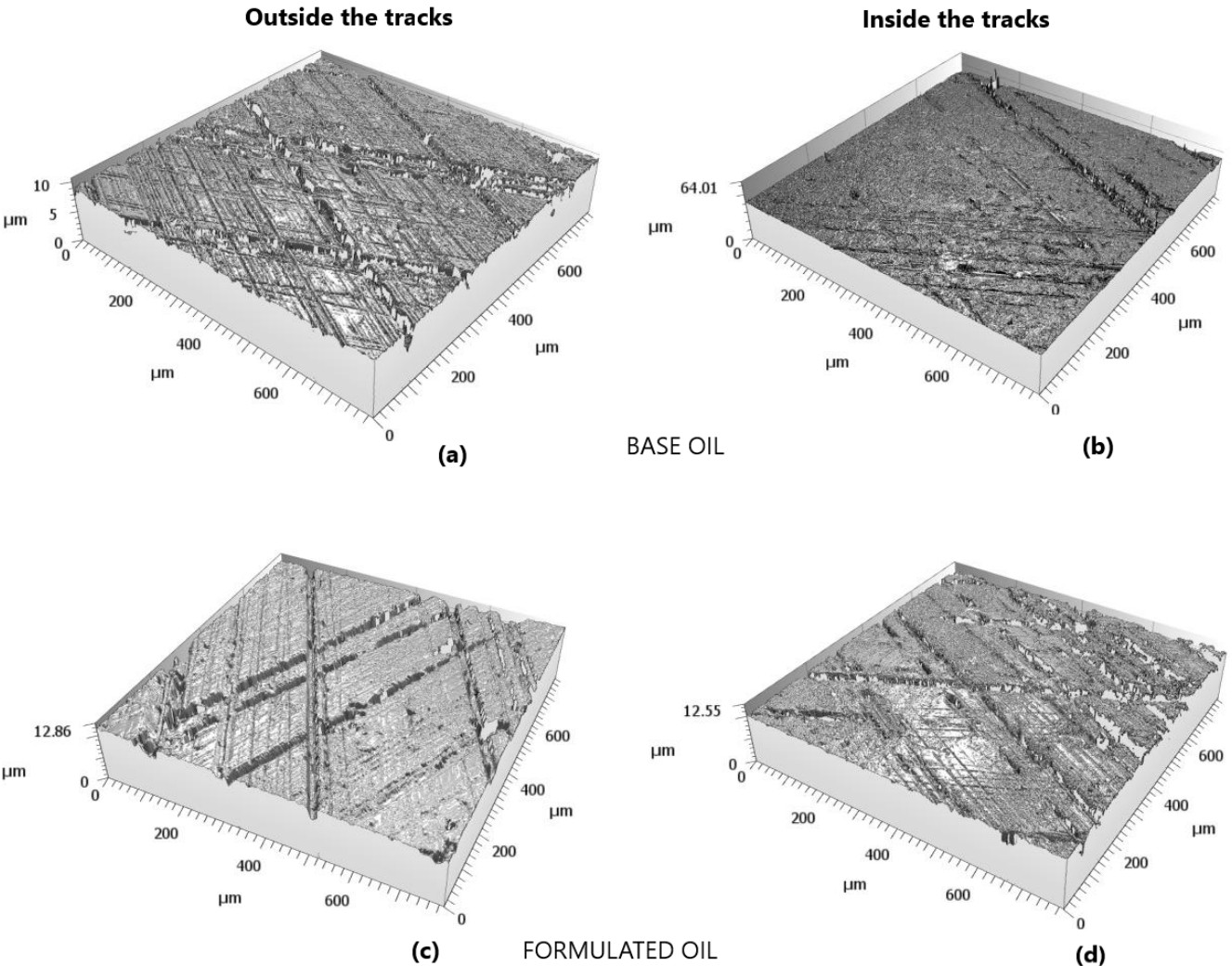

**Figure 11.** 3D maps for the honed surfaces: (**a**) base oil, outside track; (**b**) base oil, inside track; (**c**) formulated oil, outside track; (**d**) formulated oil, inside track.

SEM-EDS analysis helped to understand the morphology and composition of the worn surfaces for the different lubricant/surface finish combinations. The morphology of a honed surface before the sliding tests is shown in Figure 12a, with the typical topography of plateaus and crossed grooves, with some folded metal around the grooves (indicated by blue arrows). The results of the chemical analysis by EDS in different regions of the specimens are summarized in Figure 12d. The EDS analysis of the entire area delimited by the red rectangle 1 (shown in Figure 12a) shows a typical chemical composition of gray cast iron. Small regions near cracks (such as point 6 in Figure 12a) showed higher phosphorous content, probably corresponding to the presence of the eutectic steadite (cracks are indicated by orange arrows). After the sliding tests in the base oil (Figure 12b), the honing grooves were largely removed, as also detected by the 3D topographic maps, where only the deeper and wider grooves remained. Some chemical contrast with thin patches of darker regions covering the plateaus (e.g., points 2 and 3, one patch is highlighted by a yellow rectangle to help visualization) is shown in the backscattered electron (BSE) image. The EDS analysis shown in Figure 12d confirms a high oxygen content, indicating tribo-oxidation. For the tests with the formulated oil (Figure 12c), thicker patches of rough and dark regions appear on the plateaus, as exemplified by the area highlighted with the green rectangle. Their

chemical analysis shows higher phosphorous, zinc, and sulphur content (compare points 4 and 5), suggesting the formation of a rough and patchy ZDDP tribofilm.

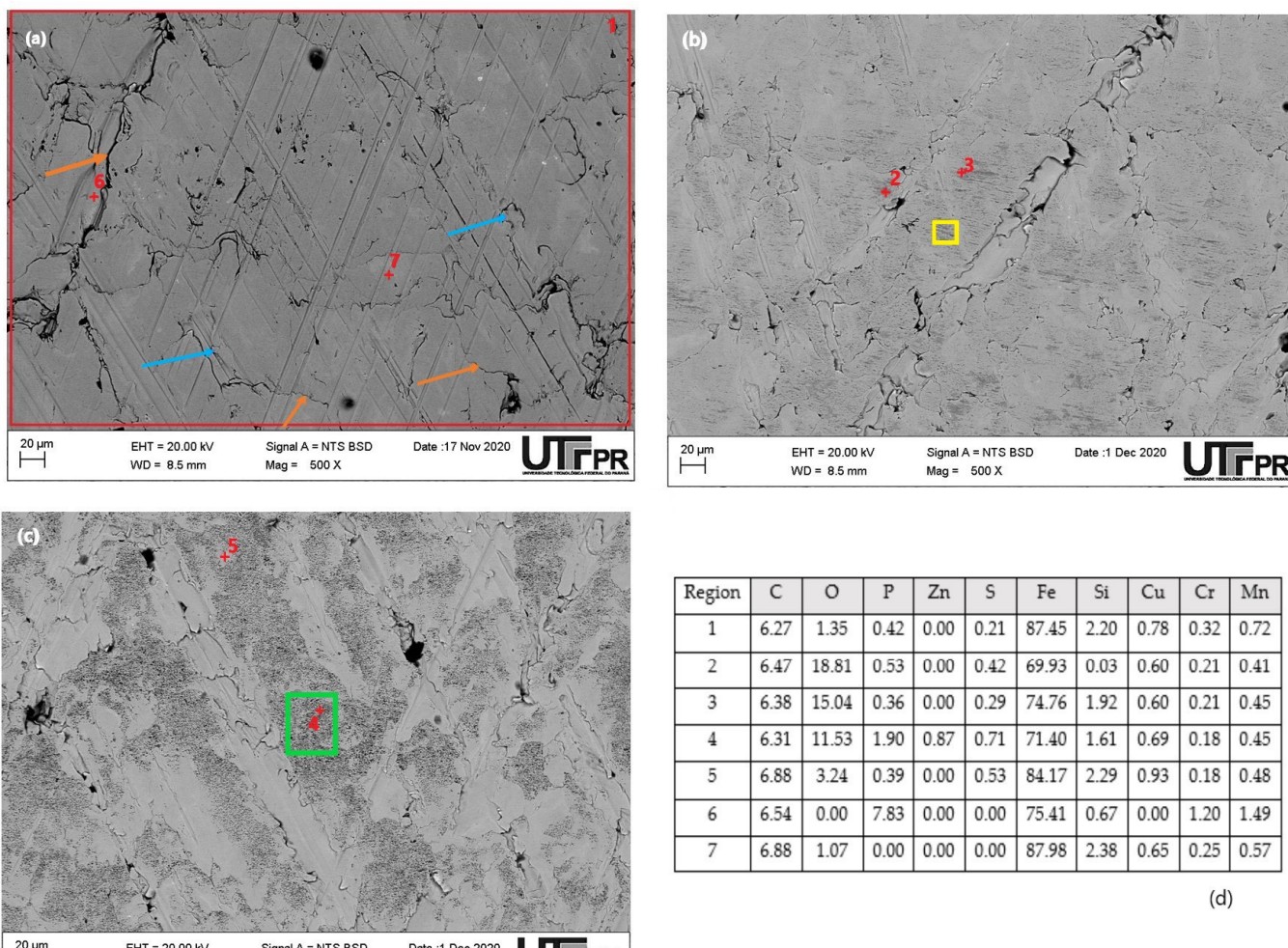

**Figure 12.** BSE SEM images of the honed surfaces: (**a**) before the tests; (**b**) after the tests in PAO4; (**c**) after the tests in formulated oil; (**d**) chemical composition by EDS of selected regions and points.

After the sliding tests in base oil with the textured liners, the surfaces appeared oxidized, as shown in Figure 13a, similarly to that observed for the honed liners. This was detected using the EDS analysis described in Figure 13c. Oxygen enrichment was detected both in the central part of the rings (point 1) and in the areas outside the pockets (2), confirming asperity contact in the central part of the rings and therefore their role in increasing the bearing capacity of the textures. The pockets became less visible and the surfaces appeared covered with a patchy tribofilm for the formulated oil (Figure 13b). The EDS analysis presented in Figure 13c confirms the enrichment with phosphorous, zinc and sulfur in all the regions analyzed (points 3 to 6), suggesting significant ZDDP tribofilm formation. The darker regions that more or less coincide with the borders of the rings showed higher carbon and oxygen content, probably originating from the MECT process (compare with Figure 7).

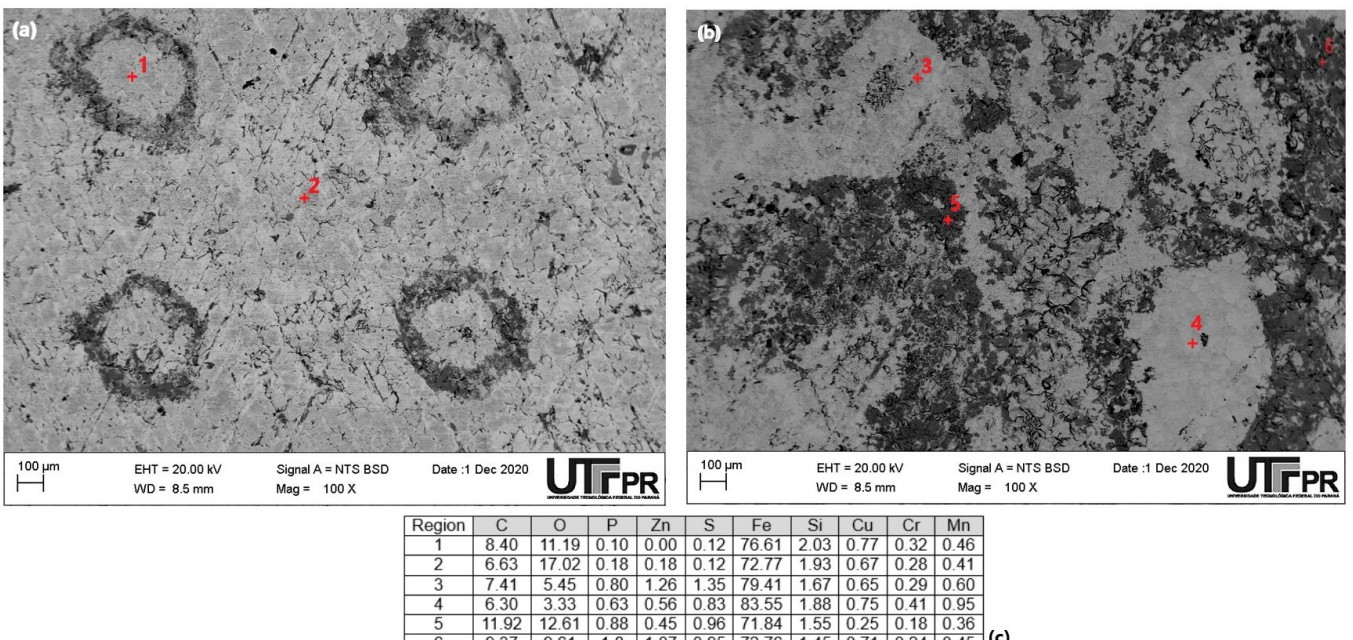

| Region | C | O | P | Zn | S | Fe | Si | Cu | Cr | Mn |
|---|---|---|---|---|---|---|---|---|---|---|
| 1 | 8.40 | 11.19 | 0.10 | 0.00 | 0.12 | 76.61 | 2.03 | 0.77 | 0.32 | 0.46 |
| 2 | 6.63 | 17.02 | 0.18 | 0.18 | 0.12 | 72.77 | 1.93 | 0.67 | 0.28 | 0.41 |
| 3 | 7.41 | 5.45 | 0.80 | 1.26 | 1.35 | 79.41 | 1.67 | 0.65 | 0.29 | 0.60 |
| 4 | 6.30 | 3.33 | 0.63 | 0.56 | 0.83 | 83.55 | 1.88 | 0.75 | 0.41 | 0.95 |
| 5 | 11.92 | 12.61 | 0.88 | 0.45 | 0.96 | 71.84 | 1.55 | 0.25 | 0.18 | 0.36 |
| 6 | 9.37 | 9.61 | 1.8 | 1.07 | 0.95 | 72.76 | 1.45 | 0.71 | 0.24 | 0.45 |

**Figure 13.** BSE SEM images after the sliding tests for the textured liners: (**a**) base oil; (**b**) formulated oil; (**c**) EDS analysis.

Some roughness parameters were used to quantify the differences in surface morphology due to wear. *Sq* values before and after the tests for both the base oil and the formulated lubricant are compared in Figure 14 to account for the average height of the asperities. For the honed surfaces, the wear tests caused a slight increase in *Sq*, which was more prominent for the base oil. However, significant scattering of the results was observed, particularly for the base oil, probably because on the one hand wear removed some of the honed grooves (tendency to reduce average height), but on the other hand roughened the plateaus (tendency to increase average height). For the textured liners, *Sq* reduced after the tests in base oil, probably because the partial removal of ring-like pockets was more significant than the roughening of the plateaus, since the pockets are much larger than the honing grooves. For the formulated oil, the formation of a protective ZDDP tribofilm tends to reduce wear, but the tribofilm is rough by nature [59]. The final result on the roughness of the honed liner was a slight increase in *Sq*, but for the textured liner the increase in *Sq* was much more significant. It might be related to a more significant formation of the rough ZDDP tribofilm, as suggested by the 3D maps previously shown in Figure 10 and will be checked by analyzing other roughness parameters.

Two important parameters for analyzing the topographies consisting of receding features (valleys) in a plateaued surface are the skewness (*Ssk*) and kurtosis (*Sku*) of the height distribution curve. In the literature [60,61], when $S_{Sk}$ and $S_{ku}$ are simultaneously analyzed by plotting $S_{ku}$ as a function of $S_{sk}$, this morphological space can offer an interesting analysis, identifying groups with similar behaviors as well as morphological changes caused by wear. In Figure 15, the honed surfaces are represented by diamonds and the textured surfaces by circles, to facilitate the visualization. As expected, all surfaces showed negative skewness values, since they have a plateau–valley morphology. Surprisingly the textured surface showed the least negative values of *Ssk* (before wear), probably because the oxidation inherent to the MECT process led to the formation of a few oxidation peaks. Since those should be very easily removed during wear, the values of *Ssk* reduced substantially with wear, particularly with the formulated oil. Kurtosis values of 3 represent a Gaussian height distribution curve, so all the surfaces were highly non-gaussian with *Sku* >> 3. After the wear tests the *Sku* values tended to be lower for the formulated oil than for the base oil, probably because the few protruding features caused by the roughening of the plateaus

observed for the base oil did not occur when a protective ZDDP tribofilm is formed, even though ZDDP tribofilm is rough itself.

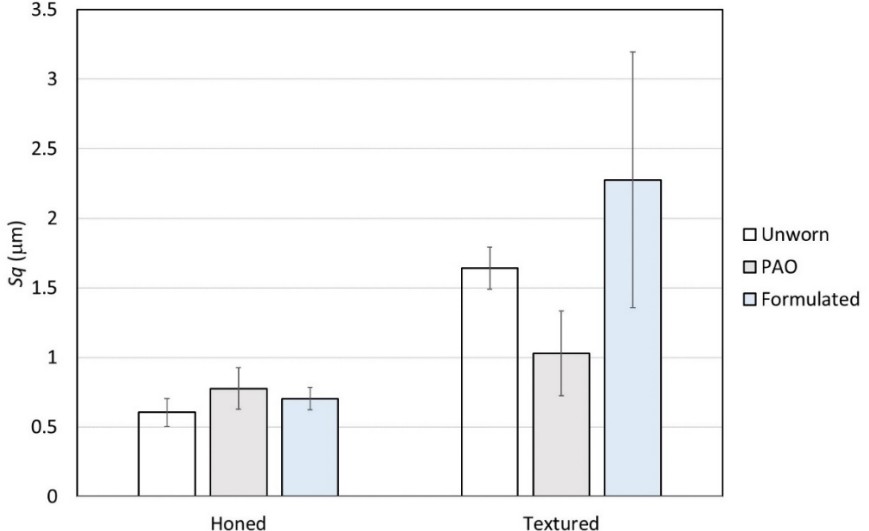

**Figure 14.** *Sq* values for the different surface/lubricant combinations.

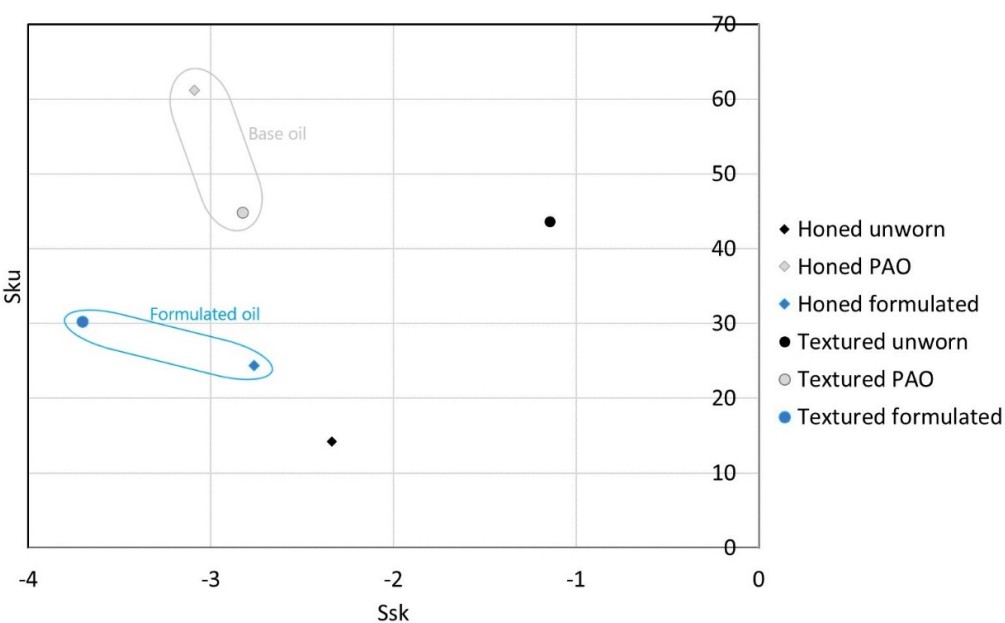

**Figure 15.** *Sku x Ssk* morphological space for the different surface/lubricant combinations.

Finally, the parameters of the *k* family (*Sk*, *Spk*, *Svk*) were used to quantify the wear of the liners. The variation of the *k* parameters after the tests (*var*) in relation to the unworn condition ((*initial − final*)/*initial*·100%), so that negative *var* values indicate an increase in the *k* parameter, are presented in Table 5. For the tests with the base oil, which does not contain boundary additives, the *Spk* value (associated with the highest asperities) of the honed liner reduced only slightly, probably because the removal of the highest peaks was balanced by the light roughening of the plateaus observed in the SEM images. The reduction in the *Spk* values was much higher (65%) for the textured liner. This probably occurred for two reasons: surfaces textured by MECT show some oxidation inherent to the process, which is easily removed, but also produces some roughening outside the pockets [32], which should be smoothened with wear. *Spk* of the honed liner increased (28%) for the formulated oil, which could be traced back to the growth of a rough tribofilm,

probably a ZDDP anti-wear tribofilm [59]. Contrarily, for the textured liner *Spk* reduced, probably due to the competition of two phenomena: on the one hand the growth of a tribofilm that was very uniformly distributed over the surface (as shown in the SEM images) covered the roughness between the pockets resulting from MECT, but on the other hand the tribofilm is rough itself. This is supported by the fact that the reduction in *Spk* was lower than for the base oil (38% compared with 65%). Another interesting fact was that with the formulated oil *Sk* and *Svk* reduced substantially for both liners, suggesting that the protective ZDDP tribofilm is well distributed over the entire surface, increasing the overall load bearing capacity, despite its rough nature. The reduction in *Sk* and *Svk* was larger for the textured liner when compared with the honed liner.

**Table 5.** Variation of the *k* parameters with the wear tests.

| Liner | Condition | *Spk* (μm) | *Sk* (μm) | *Svk* (μm) | Variation of the *k* Parameters (*var*) [(*initial* − *final*)/*initial*]·100% | | |
|---|---|---|---|---|---|---|---|
| | | | | | $Spk_{var}$ | $Sk_{var}$ | $Svk_{var}$ |
| Honed | Unworn | 0.31 | 0.68 | 1.37 | | | |
| | PAO (A) | 0.30 | 0.60 | 1.93 | 2.5% | 11.4% | −40.7% |
| | Formulated (B) | 0.40 | 0.91 | 1.82 | −28.2% | −34.5% | −33.1% |
| Textured | Unworn | 2.38 | 0.99 | 3.04 | | | |
| | PAO (C) | 0.82 | 1.14 | 2.48 | 65.4% | −15.0% | 18.7% |
| | Formulated (D) | 1.47 | 2.60 | 5.01 | 38.2% | −162.3% | −64.7% |

Therefore, it can be said from the results above that surface texturing of the liners had some effect on the activation of the anti-wear additives present in the lubricant indeed. This is supported by the observation of the 3D maps of the worn surfaces (Figures 10 and 11), as well as by the SEM/EDS analysis of the worn surface images (Figures 12 and 13), which showed more tribofilm formation for the textured surface than for the honed surface. This should be attributed to the higher contact pressures around the pockets with the reduction of contact area. The ZDDP tribofilm only forms under boundary lubrication conditions, involving tribochemical reactions that are triggered by the high shear stresses induced by asperity contact [62], justifying the importance of surface texturing under boundary lubrication for ZDDP-containing lubricants. Considering that the ZDDP tribofilm has a patchy and rough nature [59], the quantification of the topographies using the average height of the asperities showed a larger increase in *Sq* for the textured liner than for the honed liner. The variation of the *k* values with wear, as suggested in [53], also supported the hypothesis of more ZDDP tribofilm formation for the textured liner than for the honed liner. Since the growth of the ZDDP tribofilm tends to increase friction [41,44,63], friction was slightly higher for the textured liner than for the honed liner when tested with the formulated oil. Since the liquid lubricant film and the ZDDP tribofilm protect the surfaces, wear of both liners was not mensurable. The analysis of the roughness parameters before and after wear showed very little damage to both liners, but it is believed that in the long run wear should be lower for the textured liner than for the honed liner due to a more uniform protective ZDDP tribofilm. This hypothesis should be investigated in future works with longer duration tests.

## 4. Conclusions

This work investigated the interaction between surface texturing and tribofilm formation from lubricant additives using reciprocating sliding tests of segments of commercial rings and liners from the piston-liner system. For comparison, tests were also carried out with a base oil.

The reference tests with the base oil showed typical behavior when testing the honed liners, where friction was substantially lower than for the textured liner. The increased roughness conferred by the MECT technique was responsible for the higher and unstable friction. SEM/EDS showed removal of the highest asperities and tribo-oxidation for both

liners. The analysis of the topographic parameters showed a reduction in the average height of the asperities after the sliding tests for the textured liner, probably due to some reduction of the depth of the texture pockets with wear. The honed liner became rougher after wear, showing surface damage on the plateau regions.

3D topographic maps showed negligible wear and substantial tribofilm formation for the tests with the formulated oil, which appeared more intense for the textured liner. SEM/EDS analysis confirmed that a tribofilm containing P, Zn and S covered the surfaces very uniformly for the textured liner, although it had a patchy nature.

The quantification of topographic changes with wear using the parameters *Sq*, *Ssk*, *Sku*, *Spk*, *Sk* and *Svk* supported the argument of more tribofilm formation for the textured liner than for the honed liner in the presence of a formulated lubricant. Particular insight was given by the analysis of the morpohological space *Sku* × *Ssk*, as well as by the reduction in *Spk* values for the textured liner, suggesting the formation of a more uniform tribofilm covering the worn surface when compared with the honed liner.

Friction values for the tests with formulated oil were slightly higher for the textured liners than for the honed liners, which should be attributable to the rough nature of the ZDDP tribofilm. However, it is expected that when the liner is used for long periods, the more pronounced tribofilm should protect it more effectively against wear, increasing the component´s life. This hypothesis should be investigated in future works, particularly by running either longer tests or high-frequency, low average speed tests.

The used of advanced surface analysis techniques such as XPS, XANES and AFM in futures works should help understand in more detail this synergy between surface texturing and tribofilm formation, particularly for different texture geometries. The choice of surface textures needs to be guided by simulation of the stresses developed in the contact during rubbing.

**Author Contributions:** Conceptualization, H.L.C.; methodology, L.C.D., A.A.O.F.V., G.P. and H.L.C.; formal analysis, L.C.D., G.P. and H.L.C.; investigation, L.C.D.; resources, H.L.C. and G.P.; writing—original draft preparation, H.L.C.; writing—review and editing, H.L.C. All authors have read and agreed to the published version of the manuscript.

**Funding:** This research was funded by Fapergs (Brazil), grant number 19/2551-0001849-5 and CNPq (Brazil), grant number 305453/2017-3. Moreover, L.C.D. acknowledges financial support from Capes (Brazil).

**Data Availability Statement:** Not applicable.

**Acknowledgments:** The authors acknowledge donation of lubricants by Idemitsu. The authors also thank the Multi-User Center for Materials Characterization (CMCM) of the UTFPR for the SEM-EDS and interferometry analyses.

**Conflicts of Interest:** The authors declare no conflict of interest.

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
