# Peer review of "ZDDP Tribofilm Formation from a Formulated Oil on Textured Cylinder Liners"

_lubricants, doi:10.3390/lubricants10060118_

Round 1

Reviewer 1 Report

The present manuscript addresses the synergetic effects of surface texturing via Maskless Electrochemical Texturing (MECT) and ZDDP additives for engine oils employed for the cylinder liner contact. It was reported that the texturing changed the chemical composition of the grey cast iron, exposing the carbon in the metal matrix. The tribological performance was evaluated using a base and a commercially formulated oil. Compared to a commercially honed liners, the textured surfaces showed higher friction but also more pronounced ZDDP tribofilm formation.

The overall topic of the complex interplay between surface textures and lubricant additives is highly interesting for the scientific community. The manuscript is well written and scientifically sound. However, I have some comments and the following points should be addressed prior to further consideration:

  1. The ZDDP aspect could be included in the title.
  2. The language of the manuscript should be checked, especially the use of adverbs instead of adjectives, e.g. it should be "commercially honed" instead of "commercial honed" etc. There are also some misplaced line breaks that make reading difficult.
  3. "Error! Reference source not found" appears multiple times starting from section 2. There are multiple figure 1, 2, and 3. This makes it very hard or impossible to check the statements.
  4. Some of the figures are too small to be readable, especially Fig. 3, 7, 8. Also, axis labels should be added and more readable colors should be used (e.g. light grey on white background is not recommended).
  5. This is a subjective comment: I prefer when references to Figures or Tables are formulated passively ("something is shown in Figure 1" instead of "Figure 1 shows something"), because the authors show something with the item and the Figure/Table itself cannot perform any active actions. The authors can feel free to consider this or not.
  6. How were the dimensions (not process parameters) of the produced textures/dimples chosen?
  7. The relevant review articles regarding the state-of-the-art of surface texturing (especially regarding hydrodynamic contacts such as the cylinder liner) as well as ZDDP tribofilm formation should be mentioned.
  8. I believe Fig. 5 can be omitted.
  9. More information on the material characterization in terms of SEM and EDS should be provided.
  10. Reference for eq. 1 should be provided and it should be explained hy and how it was selected. Also, which values were selected for sigma in table 4 in the case of the textured surfaces, the values for the area without or with dimple? Moreover, it should be emphasized that the calculation of the minimum lubricant gap considers smooth surfaces only while the honing pattern as well as the textures will certainly influence the film height. Thus, the lambda values might be larger than provided. This should at least be addressed in the discussion.
  11. The authors attribute the higher ZDDP formation to the higher pressings. Are these of a hydrodynamic nature or due to the smaller contact area in the region of boundary friction?
  12. Compared to a commercially honed liners, the textured surfaces showed higher friction but also more pronounced ZDDP tribofilm formation. Does this also correspond to lower wear or not? This is not yet really clear from the results.

Reviewer 2 Report

This paper presents an interesting study on the tribofilm formation from a formulated oil on textured cylinder 2 liners. In general, the paper is very organized and contains sufficient data and analyses. It can be accepted after addressing the following issues.

  1. The introduction part is very short to summarize key advances achieved in the fields of green machining. So please add more literature survey in recent 2-3 years in this section.
  2. Please highlight the novelty of conducting the current study as there are a lot of studies dealing with the topics of textured cylinder liners.
  3. How did the authors calcite the COF of the contact surface? The underlying mechanisms controlling tribofilm formation should be carefully elaborated.
  4. 1 is shown with a poor resolution and contains noise. Please remove the Nosie signals from the figure.
  5. How many repetitions of tests were conducted to get reliable results?
  6. That authors have to provide a detailed indicator to quantify the wear extents during the frictional tests.
  7. The conclusions part is rather short in length, and more details should be added.
  8. The format of the paper should be carefully corrected according to the journal’s template.

Reviewer 3 Report

  1. In Fig.13(a-c) the elements, pores and agglomerations need to be show with the arrow marks.
  2. Mention EDAX images to show the presence of various elements in the selected samples.
  3. Novelty of the work need to be added at the end of the introduction.
  4. Future scope of work can be included at the end of the conclusions.
  5. Cite latest articles related to wear and lubrication example
  • Surya, M. S., & Gugulothu, S. K. (2022). Fabrication, mechanical and wear characterization of silicon carbide reinforced Aluminium 7075 metal matrix composite. Silicon, 14(5), 2023-2032.
  • Sridhar, A., & Lakshmi, K. P. (2021). Evaluation of mechanical and wear properties of aluminum 7075 alloy hybrid nanocomposites with the additions of SiC/Graphite. Materials Today: Proceedings44, 2653-2657.
  • Kumar, V. N., Kishore Nath, N., & Ramesh Babu, P. (2020). Effect of reinforcement and fabrication of Al6061 nanosilica composite prepared using single-and two-step methods. Advances in Materials and Processing Technologies, 1-20.

Round 2

Reviewer 1 Report

Thank you for addressing my comments. I'm satisfied with the changes and recommend acceptance of the article in its present form.

Reviewer 2 Report

The paper can be accepted now.

Reviewer 3 Report

Rejected